# Combined Effect of Chitosan Coating and Laurel Essential Oil (*Laurus nobilis*) on the Microbiological, Chemical, and Sensory Attributes of Water Buffalo Meat

**DOI:** 10.3390/foods11111664

**Published:** 2022-06-06

**Authors:** Lydia K. Karakosta, Kornilia A. Vatavali, Ioanna S. Kosma, Anastasia V. Badeka, Michael G. Kontominas

**Affiliations:** Laboratory of Food Chemistry, Department of Chemistry, University of Ioannina, 45110 Ioannina, Greece; lydiakarak@yahoo.gr (L.K.K.); kvatavali26@gmail.com (K.A.V.); i.kosma@uoi.gr (I.S.K.)

**Keywords:** water buffalo meat, preservation, shelf-life extension, chitosan, laurel essential oil

## Abstract

The combined effect of chitosan coating (CHI) and laurel essential oil (LEO) on the shelf-life extension of water buffalo meat stored under aerobic packaging conditions at 4 °C was investigated. Microbiological, physicochemical, and sensory attributes were monitored over an 18-day storage period. Microbiological data indicated that the (CHI) coating along with (LEO) was the most efficient among treatments in reducing populations of bacteria by 3.2 log cfu/g on day 6 of storage (*p* < 0.05). pH values of meat varied between 6.04 and 6.21, while thiobarbituric acid (TBA) values were equal to or less than 2.12 mg malondialdehyde/kg throughout storage. The colour parameter L* and a* values decreased, while b* values increased during storage (*p* < 0.05). Taste proved to be a more sensitive sensory attribute than odour. Based on sensory and microbiological data, product shelf life was approximately 5–6 days for control samples, 7–8 days for samples treated with (LEO), 12 days for samples treated with (CHI), and 13–14 days for samples treated with (CHI + LEO).

## 1. Introduction

There are approximately 204 million water buffaloes worldwide, 95% of which are found in Asia (India, Pakistan, China, and Nepal), 2% in Egypt, 2% in North and South America, and less than 1% in Europe and Australia are bred for their milk and meat [1]. The world buffalo meat production in 2019 was 4.3 million tons, mainly attributable to Asia. In Greece, there are currently 3200 water buffaloes, 80% of which are located in Serres (northern Greece); the rest (20%) are located in Komotini (north eastern Greece), Florina, and Preveza (north western Greece and Fthiotida (central Greece) and bred mainly for their meat and a lot less for their milk [2]. Buffalo meat production mainly derives from older animals at the end of their productive life and only to a small extent from young animals. This leads to meat that often do not meet consumer expectations mainly due to its dark colour and reduced tenderness due to high fiber content [3].

On the other hand, buffalo meat’s beneficial properties such as reduced fat and cholesterol content have led certain researchers to define it as “the healthiest meat among red meats for human consumption” [4]. TenderBuff^®^ (Winnellie, NT, Australia) has recently marketed a quality-assured product based on pricing buffalo meat on an individual animal basis meeting predefined specifications [5]. Likewise, Italy has created a trademark to valorise buffalo meat called Sapore di Campania (“taste of Campania”), from the name of the Campania region where buffalo farming is concentrated. The product should meet specific quality requirements for the content of fat, protein, cholesterol, and iron in order to be placed on the market [6].

Water buffalo meat is higher in muscle and lower in fat, marbling, and bone than beef. It is tougher than beef and has a lower cholesterol content. It contains more protein than the meat of other mammals and has a higher proportion of lysine. Likewise, the concentration of necessary amino acids contained in skeletal muscles is higher in buffalo than in beef. It is a rich source of numerous vitamins and minerals, high in B12, B6, potassium, iron, zinc, copper, and selenium, while lower in sodium and calories [7,8]. It also contains a greater concentration of conjugated linoleic acid (1.83 mg/g fatty acid methyl esters) compared to meat from zebu-type cattle [9].

Due to the specific composition and pH, fresh meat is highly susceptible to microbial growth and chemical deterioration including lipid oxidation, both leading to rapid spoilage and loss of desirable sensory attributes such as flavour and odour [10]. The main bacteria causing spoilage of refrigerated meat are *Pseudomonas* spp., *Brochothrix thermosphacta*, *Carnobacterium* spp., Enterobacteriaceae, *Lactobacillus* spp., and *Leuconostoc* spp. Microbiological growth and sensory defects result in a decrease in meat shelf life which varies from a few days up to several months depending on the method of preservation [8,10].

Contemporary research has focused on the use of natural preservatives as a result of consumers’ concern regarding the safety of foods containing synthetic preservatives along with the economic impact of spoiled foods. Natural preservatives such as chitosan, essential oils, bacteriocins, organic acids, and modified atmosphere packaging are used for delaying microbial spoilage and controlling fat oxidation [11,12,13,14].

Bay laurel (*Laurus nobilis*) is a native species of the Mediterranean region with moderate and subtropical climate belonging to the *Lauraceae* family [15,16]. The dried leaves and essential oil are used extensively as a spice and flavouring agent in the culinary and food industry. Their flavor suits mostly cooked red meat and poultry according to Mediterranean recipes [17]. Laurel leaves and essential oil also have preservative, antioxidant, and antimicrobial properties [15,16,18]. Generally, the yield and composition of the oil varies between 1–5% depending on the part of the plant used to isolate it, geographical origin, the variety, and the harvest season [19]. Constituents responsible for its antimicrobial and antioxidant activity include: monoterpene hydrocarbons (α- and β-pinene, sabinene), oxygenated monoterpene hydrocarbons (1,8-cineol, α-terpinyl acetate, linalool, α,γ-terpineol), and aromatic compounds (eugenol and methyl-eugenol) that can exhibit preservative properties through different mechanisms [20].

Chitosan [b-(1,4)-2-amino-2-deoxy-D-glucopyranose] is produced from chitin deriving from crustacean shells through deacetylation. It is both a biodegradable and biocompatible polymer, the second most abundant in nature next to cellulose. Due to a number of functional properties, including its non-toxic nature, its antimicrobial and antioxidant activity and its ability to form protective films, chitosan has attracted attention as a potential food preservative of natural origin [21,22].

Chitosan has been reported to inhibit the growth of a wide variety of Gram-positive and Gram-negative bacteria and fungi [23,24]. Potential applications of chitosan as a biopreservative have been investigated in various meat products against various food spoilage and pathogenic microorganisms [12,25,26,27].

To the best of our knowledge, the use of (LEO) individually or in combination with chitosan has not been investigated in freshwater buffalo meat preservation, hence the novelty of the present study. Thus, the aim of this study was to determine the effect of chitosan coating and (LEO), applied individually or in combination, on the microbiological, chemical, and sensory properties of water buffalo meat, packaged aerobically during refrigerated storage.

## 2. Materials and Methods

### 2.1. Preparation of Coating Solutions and Treatment of Meat Samples

Fresh meat (*Longissimus dorsi*) from young male 20–24-month-old water buffaloes was obtained from a commercial water buffalo farm in Preveza, Epirus, (NW Greece), in July 2020, after slaughter and ageing for 24 h at 4 ± 1 °C. It was placed in polystyrene boxes, in ice and transferred to the laboratory within 1 h. After specific treatment, the meat samples were stored under refrigeration (4 ± 1 °C).

Food grade chitosan from shrimp shells, (molecular weight: 100 kDa, degree of deacetylation: 95%) in the form of a powder, was provided by Primex ehf (Siglufjordur, Iceland). A 1.25% (*w/v*) (CHI) solution was prepared in 1% *v/v* acetic acid (AA). To achieve complete dissolution of chitosan, the (AA)/(CHI) suspension was left under stirring at room temperature for 24 h on a magnetic plate with the aid of a magnetic stirring bar. Glycerol (0.75 mL/g of chitosan) was added as a plasticiser. A 1.25% (*v/v*) (LEO) solution in water was prepared with the addition of 2% *v/v* dimethyl sulfoxide (DMSO), as an emulsifier. (LEO) was donated by the Department of Agriculture, University of Ioannina, Arta, Greece, where it was prepared by steam distillation of bay leaves harvested locally according to the official method of the European Pharmacopoeia [28].

Based on the volume change of the (CHI) and (LEO) solution before and after the dipping of the meat, a concentration of 0.1% in either (CHI) or (LEO) was calculated. In preliminary experiments, it was shown that concentrations of (LEO) higher than 0.1% gave a strong caustic taste to meat. Finally, a solution containing both (CHI) and (LEO) was prepared as described above.

For each sample, approximately 150 g of water buffalo meat was aseptically portioned into smaller pieces of about 30 g and put on a skewer. The following lots of samples were prepared: the first lot comprised the control samples (C), dipped in water plus DMSO and glycerol for 2 min and then allowed to drain for 5 min; the second lot comprised samples dipped in a 1% *v/v* (AA) solution; the third lot comprised samples dipped in a 1.25% *w/v* (CHI) solution; the fourth lot comprised samples dipped in a 1.25% *v/v* (LEO) solution; the fifth lot comprised samples dipped in a 1.25% (CHI) plus 1.25% (LEO) solution (CHI + LEO).

After dipping and draining, the samples were packaged in low density polyethylene (LDPE- 65 μm in thickness) pouches, heat- sealed using a BOSS model No48 thermal sealer (BOSS, Bad Humburg, Germany) and stored under refrigeration at 4 ± 1 °C. The oxygen permeability and water vapour transmission rate of the LDPE film used were 5000 cm^3^/m^2^/24 h/atm and 24 cm^3^/m^2^/24 h/atm, measured using the Oxtran 2/20 and Permatran 3/31 permeability testers respectively (Mocon, Co., North Brooklyn Park, MN, USA). Sampling was carried out at predetermined time intervals, on 0, 3, 6, 9, 12, 15, and 18 days of storage.

In order to determine the optimum concentration of each antimicrobial agent applied to the water buffalo meat samples, preliminary sensory evaluation and total viable count (TVC) analyses were run.

### 2.2. Physicochemical Analysis

The pH was recorded using a Delta OHM, model HD 345, pH-meter at ambient temperature (Delta OHM s.r.l. Caselle di Selvazzano, Selvazzano Dentro, Italy). A total of 10 g of buffalo meat muscle was homogenised thoroughly with 90 mL of distilled water and the homogenate was used for pH determination. Colour determination was carried out on meat samples using a Hunter Lab, Model D25 L, optical sensor colourimeter (Hunter Associates Laboratory, Reston, VA, USA) as described by Mexis et al. [29]. TBA was determined according to the method of Goulas and Kontominas [30]. Briefly, the meat sample was homogenised with a trichloroacetic acid (TCA) and butylatedhydroxyltoluene (BHT) solutions and centrifuged. The sample was filtered and TBA solution was added to the filtrate. The sample was then incubated in a water bath at 70 °C for 90 min to form a pink complex. The absorbance of the complex was then measured at λ = 532 nm against a blank using a malondialdehyde standard curve. TBA content was expressed as mg of MDA/kg meat muscle.

The identification of the volatile compounds in (LEO) was carried out according to Stefanova et al. [18] in a 0.1% solution of EO prepared in hexane. An Agilent 7890A series gas chromatograph equipped with an Agilent 5975C inert XL MSD mass selective detector (Wilmington, DE, USA) was used. The column was HP5-MS (30 m × 250 μm × 0.25 μm), temperature program: 35 °C/3 min, 5 °C/min to 250 °C for 3 min. Helium as the carrier gas, flow rate: 1 mL/min., 30:1 split ratio. The identification of the volatile compounds was made by comparison of their mass spectra to those of the Wiley Library.

### 2.3. Microbiological Analysis

For determination of microbial counts, 25 gr of meat samples were transferred aseptically into individual stomacher bags (Seward Medical, Wothing, UK) containing 225 mL of Buffered Peptone Water solution (BPW, 0.1%) and homogenised for 60 s using a Lab Blender 400, Stomacher (Seward Medical, Worthing, UK), at room temperature. For each sample, further serial decimal dilutions were prepared in BPW solution (0.1%). The amount of 0.1 mL for these serial dilutions of meat homogenates was spread on the surface of agar plates. TVC, *Pseudomonas* spp., *Brochothrix thermosphacta*, Lactic acid bacteria (LAB), and Enterobacteriaceae were enumerated according to APHA [31].

### 2.4. Sensory Evaluation

After each sampling, meat samples were frozen (−20 °C) until sensory evaluation. Frozen meat samples were thawed, and steam-cooked on a metal rack for approximately 30 min. The attributes (odour and taste) of cooked buffalo meat on each sampling day were evaluated by a panel of 51 untrained judges, graduate students, and faculty of the Laboratory of Food Chemistry, University of Ioannina, according to Chounou et al. [12]. The panel consisted of 28 females and 23 males in the 22–60 age group who consume meat on a regular basis. The scoring scale was 1–5, where 5 corresponded to the most liked sample and 1 corresponded to the least liked sample; a score of 3 was the lower acceptability limit. Minimum information was given to panelists with regard to the odour and taste of cooked meat. Tenderness of cooked buffalo meat was not evaluated as the meat used originated from young male 20–24-month-old water buffaloes [8].

### 2.5. Statistical Analysis

Experiments were replicated twice on different occasions with different meat samples. Analyses were run in triplicate for each replicate (*n* = 2 × 3 = 6). Data were subjected to a two-way analysis of variance (ANOVA) using the software of Minitab 16. Microbiological data were transformed into logarithms of the number of colony forming units (cfu/g). More specifically, a two-way ANOVA was carried out using the software SPSS 16 for Windows to investigate the effect of the independent variables (treatments and time) used in the study on dependent variables (TVC, Pseudomonads, *Br. thermosphacta*, Enterobacteriaceae, LAB, odour, taste, colour parameters L*, a* and b*, pH, TBA). In addition, the possible interaction between parameter time and treatments was investigated. The model applied in each case was in the form:Y*_ijk_* = μ + α*_i_* + β_j_ + (αβ)*_ij_* + ε*_ijk_*
*i* = 0, 3, 6, 9, 12, 15, 18 *j* = 1, …, 5 *k* = 1, …, 6
where Y*_ijk_* is the log of the number of cfu/g of meat sample *k*, *k* for dependent variable, stored for *i* days under treatment *j*; α*_i_* is the effect of storage time *i* on the value of two independent variables; β_j_ is the effect of treatment j; (αβ)*_ij_* is the interaction between time *i* and treatment *j*; μ is the true mean value of all the observations; and ε*_ijk_* is the random error.

Means and standard errors were calculated. Significance was defined at *p* = 0.05 and when F-values were significant at *p* < 0.05 level, mean differences were separated by the least significant difference (LSD) procedure.

## 3. Results and Discussion

### 3.1. Physicochemical Changes

#### 3.1.1. Composition of Laurel Essential Oil

The composition of (LEO) is shown in Table 1. Τhirty-nine volatile compounds were identified and semi-quantified. The main constituents of (LEO) in decreasing concentration order are: 1,8- cineole (22.44%), α-terpinenyl acetate (16.40%), sabinene (6.90%), α- pinene (5.79%), α-terpineol (5.40%), 2- β- pipene (4.93%), methyl eugenol (4.22%), 4-terpineol (4.36%), eugenol (3.25%), L-linalool (2.99%), and dl- limonene (2.69%). A number of these compounds have been reported to possess antimicrobial properties [15,16,32]. A similar composition for (LEO) has been reported by Marzouki et al. [33], da Silveira et al. [34], Ordoudi et al. [35], and Stefanova et al. [18].

#### 3.1.2. PH

Changes in the pH values during storage under aerobic packaging conditions are shown in Figure 1a. The initial pH of buffalo meat on day 0 was 6.04. During storage, pH of all the samples increased (*p* < 0.05) until day 6–7 and after that decreased. Lower pH values were recorded for (AA) treated samples (6.14 on day 6 of storage). This may be attributed to the dipping of the meat in an acidic solution. Although the pH of the samples coated with (CHI), (LEO), and (CHI + LEO) also increased during the first days of storage, it was consistently lower than that of the control samples, with statistically significant differences (*p* < 0.05) between the two until day 9 of storage. At pH values lower than 6.4, chitosan acts as an unbranched cationic biopolymer, which negatively affects charged materials such as bacteria, fungi, etc. Such a pH change may be attributed to two opposing phenomena: (i) as stated above, the production of lactic acid by LAB resulting in a decrease in pH values and (ii) protein breakdown for the production of alkaline compounds (NH_3_), resulting in increased pH values [36]. Until day 6 of storage, the second mechanism prevailed while the LAB population was still low. During the later stages of storage, the contribution of LAB was more dominant, resulting to the net reduction in pH. Studies on the preservation of raw buffalo meat are rather limited. Results of the present study regarding pH are in good agreement with those of El-Saadony et al. [37] used bioactive peptides (Alcalase-red kidney bean hydrolysate (RBAH) and 11S pea globulin (11SGP)) in the form of coatings to extend the shelf life of raw buffalo meat. The pH values significantly (*p* < 0.05) increased during the storage period from pH 6 to 7.2 in control samples, this increment reduced with a relative decrease of about 6–14% in meat samples supplemented with RBAH and 11SGP concentrations (100, 250 and 400 mg/g). Present results are only in partial agreement with those of Chounou et al. [12], who reported statistically insignificant changes in pH of chitosan- treated (1% *w/w*) ground beef meat.

#### 3.1.3. TBA

The initial TBA value for meat (Figure 1b) was 0.76 mg MDA/kg meat and increased with storage time reaching mean values of 2.12 for the control samples, 1.33 for samples treated with (AA), 1.37 for samples treated with (LEO), 0.74 and 0.69 for samples treated with (CHI), and (CHI + LEO), on day 6 of storage. Lower TBA values for (LEO) and/or (CHI)-treated samples are due to the documented antioxidant activity of both. (CHI) and (CHI + LEO) treatments resulted in sample TBA values below 1.5 mg MDA/kg throughout storage. Such values are lower or near the proposed limit of 2 mg MDA/kg meat, above which rancid off-flavours become sensorily detectible in meat products [38]. As shown in Figure 1b, (LEO) significantly affected (*p* < 0.05) the oxidation of buffalo meat compared to the control sample. Chitosan was even more effective in reducing the MDA content of buffalo meat by 2-fold during storage (*p* < 0.0). Use of chitosan results in the formation of a thin barrier film on the surface of meat samples protecting it from the attack of oxygen. Kandeepan and Biswas [39] reported a TBA value of 0.32 mg MDA/kg for raw, untreated buffalo meat stored for 7 days in polyethylene bags at 4 °C, a value substantially lower than values reported in the present study. Differences in TBA values between the two may be related to different buffalo meat samples used and different oxygen permeability of PE bags used to package meat. Chounou et al. [12] reported that at the point of product sensory rejection, ground meat treated with 1% chitosan had a MDA content of 1 mg/kg, showing a 33% reduction as compared to control samples (1.5 mg/kg).

#### 3.1.4. Colour

Colour parameter values (L*, a*, b*) as a function of storage time are given in Figure 2a–c. L* (degree of lightness) decreased throughout storage, indicative that the colour of buffalo meat became duller. The decrease in L* values was significantly steeper in control samples compared to treated ones, i.e., on day 9; L* values for control samples were considerably lower (*p* < 0.05) than those of all treated samples. On the other hand, both (LEO) and (CHI) provided partial protection to the colour (parameter L*) of buffalo meat exhibiting higher values of meat sample lightness compared to the control. According to Vatavali et al. [40], changes in L* colour value are related to the gradual decomposition of muscle proteins leading to increased diffusion of incident light, resulting in turn to decreased L* values. In (CHI + LEO)-treated samples, L* colour values showed a slightly decreasing trend as a result of the preservative effect of (CHI) and (LEO) on meat proteins. Kahn et al. [41] reported a L* colour parameter value of 36.5 for buffalo meat and 43.5 for cattle meat. Such colour parameter values for buffalo meat are in reasonable agreement with those of the present study (40.5–38). Similar to these findings, Chounou et al. [12] reported a decreasing trend in L* values during storage of ground meat samples with those containing chitosan (1% *w/w*) (*p* < 0.05) exhibiting a lower decrease in L* values compared to the control sample. On the other hand, El-Saadony et al. [37] reported no statistically significant differences in L* values in buffalo meat treated with bioactive peptides (alcalase-red kidney bean hydrolysate and 11S pea globulin) compared to controls after 15 days of refrigerated storage.

Likewise, parameter a* values (related to degree of redness–greenness), decreased during storage (*p* < 0.05) as a result of partial loss of the red colour of meat due the oxidation of oxymyoglobin (Fe^2+^) to metmyogloobin (Fe^3+^). The same decreasing trend was shown for a* parameter values in the studies by El-Saadony et al. [37] and Tremonte et al. [42], who investigated the shelf-life extension of refrigerated water buffalo steaks upon the addition of Malpighia punicifolia (MP) extract at concentrations in the range of 0.025 and 0.05% for a period of 21 days.

Kahn et al. [41] reported a* colour parameter values were 18.2 for buffalo meat and 14.9 for cattle meat in reasonable agreement with those of 16–11 for buffalo meat in the present study.

Parameter b* values, related to yellowness, increased (*p* < 0.05) with time probably as a result of meat fat oxidation. The lower b* values were observed in samples with (CHI) probably due to the protective effect of (CHI) to oxidation. Kahn et al. [41] reported similar values for colour parameter b* equal to 11.8 for buffalo meat and 10.2 for cattle meat to those of the present study (9.2–13.5). The same increasing trend in b* colour parameter values was reported in the studies of El-Saadony et al. [37] and Tremonte et al. [42] on buffalo meat with storage time and concentration of antioxidant additive.

### 3.2. Microbiological Changes

Figure 3a–e depicts changes in microbial flora of buffalo meat, as a function of treatment and storage time. The initial TVC (Figure 3a) for fresh meat was ca. 3.8 log cfu/g (day 0 of experiment, day 1 after slaughter), indicative of good quality meat [43].

TVC reached the upper limit of 7 log cfu/g, proposed by ICMSF [43], on day 5–6 for control samples (C), day 8 for samples dipped in 1.25% *v/v* (LEO), day 9–10 for samples dipped in 1% *v/v* (AA), day 13–14 for samples dipped in 1.25% *w/v* (CHI) solution, and day 16–17 for samples dipped in 1.25% *w/v* (CHI) plus 1.25% *v/v* (LEO) solution (CHI + LEO). On day 6 of storage, the TVC was reduced by ca. 1.6, 2.4, 2.8 and 3.2 log cfu/g for the (LEO), (AA), (CHI), and (CHI + LEO) treatment respectively (*p* < 0.05).

The use of (AA) resulted in a microbiological shelf-life extension of 3–4 days, obviously due to the reduction in pH. The use of (LEO) resulted in a microbiological shelf-life extension of 2–3 days, while the use of (CHI) resulted in an extension of 7–8 days. The combination (CHI + LEO) had a significant effect on the inhibition of TVC in buffalo meat resulting in a microbiological shelf-life extension of up to 11–12 days. (CHI) was more effective than (LEO) in reducing TVC populations of water buffalo meat (*p* < 0.05).

The most probable mechanism for the antimicrobial action of chitosan involves the interaction between positively charged chitosan molecules and negatively charged residues of bacterial cell surfaces, leading to (1) the leakage of proteinaceous and other intercellular constituents, or (2) the binding of chitosan with microbial DNA resulting in the inhibition of mRNA and protein synthesis via the penetration of chitosan into the nuclei of the microorganisms [24].

An important characteristic of EOs and their components is their hydrophobic nature, enabling them to interact with the bacterial cell membrane and mitochondria, disturbing their structures and rendering them more permeable [32]. An important limitation for the use of EOs in food preservation is the strong flavour they impart to foodstuffs. For this reason EOs are used in lower concentrations in combination with other hurdle technologies [32,44].

The present results regarding TVC are in general agreement with those of El-Saadony et al. [37], who reported TVC values of 5.80, 5.50, and 4.90 log cfu/g, respectively, for the control sample, raw buffalo meat samples, samples coated with alcalase-red kidney bean hydrolysate, and samples coated with 11S pea globulin after 5 days of refrigerated storage. Likewise, the present results are also in general agreement with those of Cheong et al. [45] who reported that the dipping of beef loins in 1% chitosan solution and 5% trisodioum phosphate effectively inhibited the growth of aerobic spoilage microorganisms during storage at 10°C, while Chounou et al. [12] showed that the addition of 1% *w/w* chitosan in fresh ground meat resulted in a shelf-life extension of 1 day compared to control samples.

Pseudomonads are Gram-negative aerobic bacteria, comprising the main spoilage microorganisms of meat [10]. *Pseudomonas* spp. (Figure 3b) with an initial count of ca. 2.9 log cfu/g, followed an increasing trend throughout storage. On day 6 of storage, the end of the microbiological shelf life of buffalo meat, treatment with (LEO), (AA), (CHI), and (CHI + LEO) reduced the *Pseudomonas* spp. population by ca. 1.0, 2.0, 2.5, and 2.7 log cfu/g (*p* < 0.05) compared to the control samples.

It is noteworthy that the pseudomonads appear to be least sensitive to the action of EOs, even though various studies claim the opposite [32]. Elgayyar et al. [46] reported that oregano EO was less effective in inhibiting *Pseudomonas aeruginosa* compared to other microorganisms. Skandamis et al. [47] also found that the pseudomonads were the most resistant group to oregano oil as compared to the other spoilage flora.

Karabagias et al. [11] showed that thyme EO had a small but statistically significant (*p* < 0.05) controlling effect on the pseudomonads’ population of lamb meat. With respect to chitosan, Chounou et al. [12] showed that chitosan (1%) reduced the pseudomonads’ population by 0.6 log cfu/g on day 6 of ground meat storage.

*Brochothrix thermosphacta* is a Gram-positive facultative anaerobe consisting part of the natural microflora of fresh meat packaged either aerobically or under modified atmosphere packaging (MAP) [48]. The initial count of *B. thermosphacta* (Figure 3c) was ca. 2.9 log cfu/g. Its population was reduced by 1.4, 2.1, 2.6, and 2.9 log cfu/g by (LEO), (AA), (CHI), and (CHI + LEO), respectively, (*p* < 0.05) on day 6 of storage. Generally, Gram-positive bacteria such as LAB and B. *thermosphacta* are more susceptible to antimicrobials such as chitosan, essential oils, nisin etc., because they have a considerably different cell wall structure as compared to Gram-negative bacteria [32].

With regard to EOs, present results are in general agreement with those of Tremonte et al. [42] who reported a reduction in *B. thermosphacta* counts by 1.0 log cfu/g in buffalo meat steaks with the addition of 0.05% *v/w* hydroethanolic extract of *Malpighia punicifolia* after 6 days of storage compared to the control samples. With regard to chitosan, present results are in general agreement with those of Chounou et al. [12] who reported a reduction in 1.3 log cfu/g on day 6 of storage of ground meat after the addition of 1% *w/w* chitosan compared to the control samples.

Enterobacteriaceae, a hygiene indicator, was found by Camargo et al. [49] to a lesser extent, is also part of the microflora of fresh meat. Enterobacteriaceae (Figure 3d) had a very low initial count of ca. 1.8 log cfu/g indicative of the good hygiene condition of the buffalo meat used, and on day 6 of storage, their population was reduced by 1.3 log cfu/g by either (LEO) or (AA), 2.4 and 2.7 log cfu/g by (CHI), and (CHI + LEO) (*p* < 0.05). Chounou et al. [12] reported a reduction in Enterobacteriaceae population by 0.7 log cfu/g after the addition of chitosan (1% *w/w*) in ground beef after 6 days of storage at 4 °C. With regard to the use of EOs, present results are in agreement with those of Karabagias et al. [11] who reported a reduction of 2.8 log cfu/g in lamb meat with addition of 0.1% *v/w* thyme essential oil on day 6 of storage compared to the control samples.

LAB are Gram-positive fermentative bacteria growing both in the presence or the absence of oxygen. According to Jay et al. [10] they constitute a substantial part of the natural spoilage microflora of both aerobically and vacuum packaged meat. The initial LAB count (Figure 3e) was ca. 2.7 log cfu/g. On day 6 of storage, (AA) reduced the LAB population by 0.7 log cfu/g, (CHI) by 1.4 log cfu/g, and the combination of (CHI + LEO) by 1.8 log cfu/g (*p* < 0.05) compared to control samples (*p* < 0.05). (LEO) resulted in a small but statistically non-significant reduction in the population of LAB (*p* > 0.05). According to Karabagias et al. [11] the addition of 0.1% *v/w* of thyme EO to lamb meat resulted in a 1.1 log cfu/g reduction in the population of LAB on day 6 of storage. On the other hand, Chounou et al. [12] reported a statistically non-significant (*p* > 0.05) LAB reduction by only 0.2 log cfu/g on day 6 of ground meat storage after the addition of 1% *w/w* chitosan.

### 3.3. Sensory Analysis

The results for the sensory (odour and taste) evaluation of cooked buffalo meat are presented in Figure 4a,b, respectively. At this point, it should be mentioned that laurel oil gave a subtle flavour compatible with the flavour of cooked buffalo meat. Odour and taste scores decreased significantly (*p* < 0.05) with storage time ranging between 5 and 3.2 for odour and between 5 and 2.9 for taste. Taste proved to be a more sensitive sensory attribute than odour as the limit of acceptability was reached sooner for taste than for odour (Figure 4a vs. Figure 4b). Significant differences among different treatments were clearly observed after day 6 of storage.

The lower acceptability limit of 3 was reached for taste on day 6 for the control samples, day 7 for samples treated with (LEO), day 8 for samples treated with (AA), 12 days for samples treated with (CHI), and 13–14 days for samples treated with chitosan plus laurel essential oil (CHI + LEO). As shown by microbiological data, chitosan was substantially more effective in shelf-life extension of buffalo meat as compared to (LEO). An additive effect was shown for the (CHI + LEO) combination, extending product shelf life by 7–8 days as compared to the control samples.

Present sensory data were in reasonable agreement with microbiological data. Differences observed between the two may be due to the fact that it is the specific spoilage organisms (SSO) that cause spoilage rather than the total number of microorganisms present in the product [10].

Kandeepan et al. [50] determined the shelf life of refrigerated buffalo ground meat (keema) prepared with various flavourings and condiments. Product shelf life was 18 days with an overall acceptability score ranging from extremely acceptable to moderately acceptable. Tremonte et al. [42] investigated the shelf-life extension of refrigerated water buffalo steaks with the addition of *Malpighia punicifolia* (MP) extract at concentrations in the range of 0.0063 to 0.05%. Positive results in terms of steak colour and flavour and general arrearance were observed at a MP extract concentration of 0.025% and 0.05% up to a period of 21 days.

El-Saadony et al. [37] coated raw buffalo meat with high solubility bioactive peptides (Alcalase-red kidney bean hydrolysate (RBAH and 11S pea globulin (11SGP) in an effort to extend product shelf life and reported that the supplementation of raw meat with RBAH and 11SGP (400 µg/g) resulted in the retention of product overall acceptability for a period of 2 weeks while the respective shelf life of the control sample was only 5 days.

### 3.4. Correlation between Sensory, Sensory and Physicochemical Parameters

Pearson statistics showed positive and significant correlations between specific parameters (Table 2). What is worth mentioning is the negative correlation among sensory parameters and TVC further supporting the role of specific spoilage micro-organisms in the deterioration of meat [51].

TVC was positively correlated to the Pseudomonads, Enterobacteriaceae, *Br. thermosphacta*, LAB, and TBA, whereas the pseudomonads were positively correlated to Enterobacteriaceae LAB, *Br. thermosphacta* and TBA and negatively correlated to both odour and taste. Enterobacteriaceae were positively correlated to LAB, *Br. thermosphacta* and TBA while LAB was positively correlated to *Br. thermosphacta* and TBA but negatively correlated to odour. *Br. thermosphacta* was positively correlated to TBA. Finally, taste was negatively correlated to colour parameter b*.

## 4. Conclusions

Based primarily on microbiological (TVC upper limit of 7 log cfu/g) and sensory data (lower acceptability limit of 3), it can be concluded that the shelf life of water buffalo meat stored under refrigeration was approximately 5–6 days for control samples, 7–8 days for samples treated with laurel essential oil, 12 days for samples treated with chitosan, and 13–14 days for samples treated with both (CHI) and (LEO). Thus, the combination of (CHI) and (LEO) resulted in the longest shelf-life extension of buffalo meat equal to approximately 8 days compared to the control sample.

## Figures and Tables

**Figure 1 foods-11-01664-f001:**
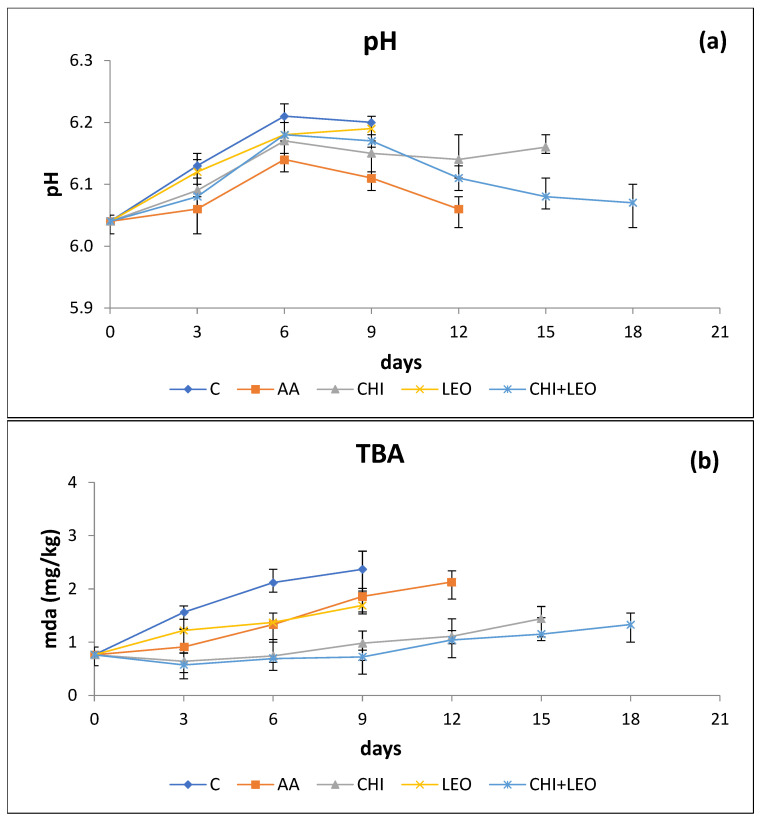
Combined effect of laurel essential oil and chitosan on (**a**) pH and (**b**) malondialdehyde content of buffalo meat as a function of storage time (C = control; AA = acetic acid; CHI = chitosan; LEO = laurel essential oil).

**Figure 2 foods-11-01664-f002:**
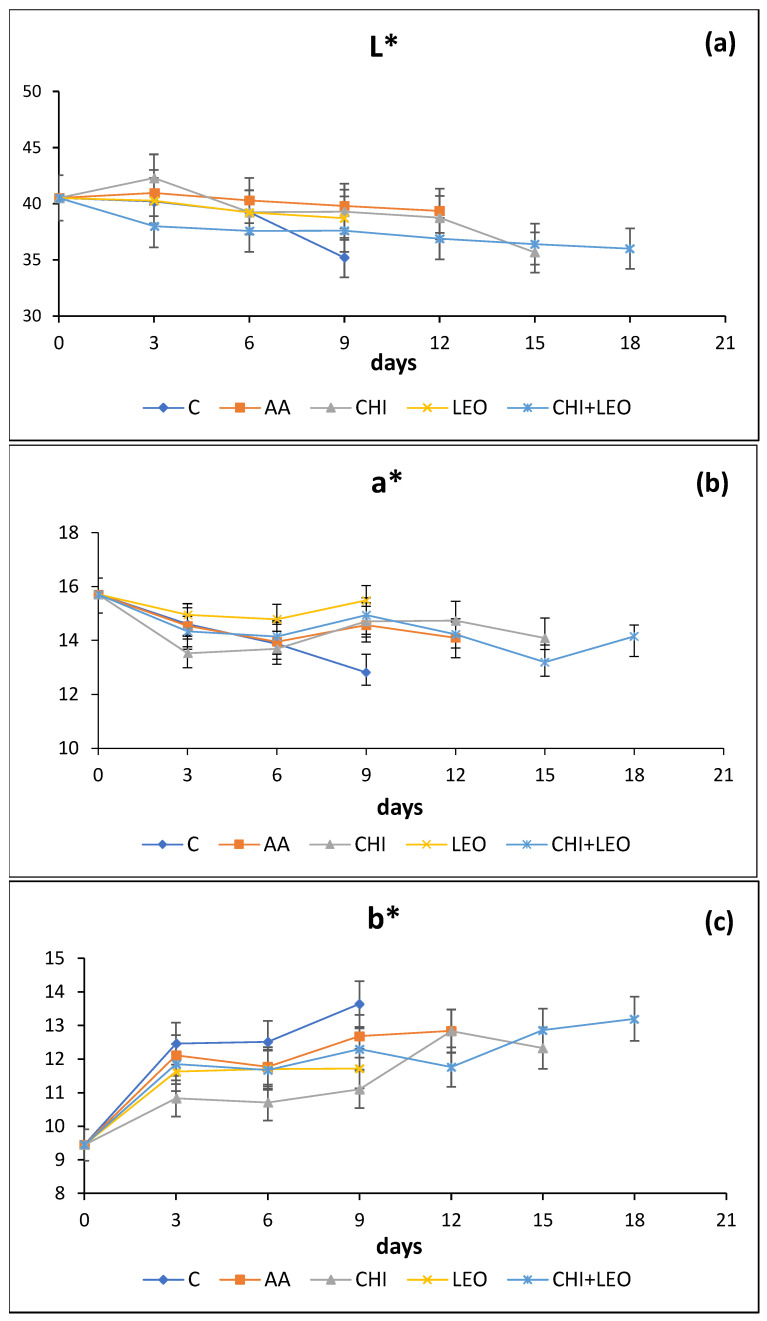
Combined effect of laurel essential oil and chitosan on (**a**) colour parameter L*, (**b**) colour parameter a*, and (**c**) colour parameter b* of buffalo meat as a function of storage time (C = control; AA = acetic acid; CHI = chitosan; LEO = laurel essential oil).

**Figure 3 foods-11-01664-f003:**
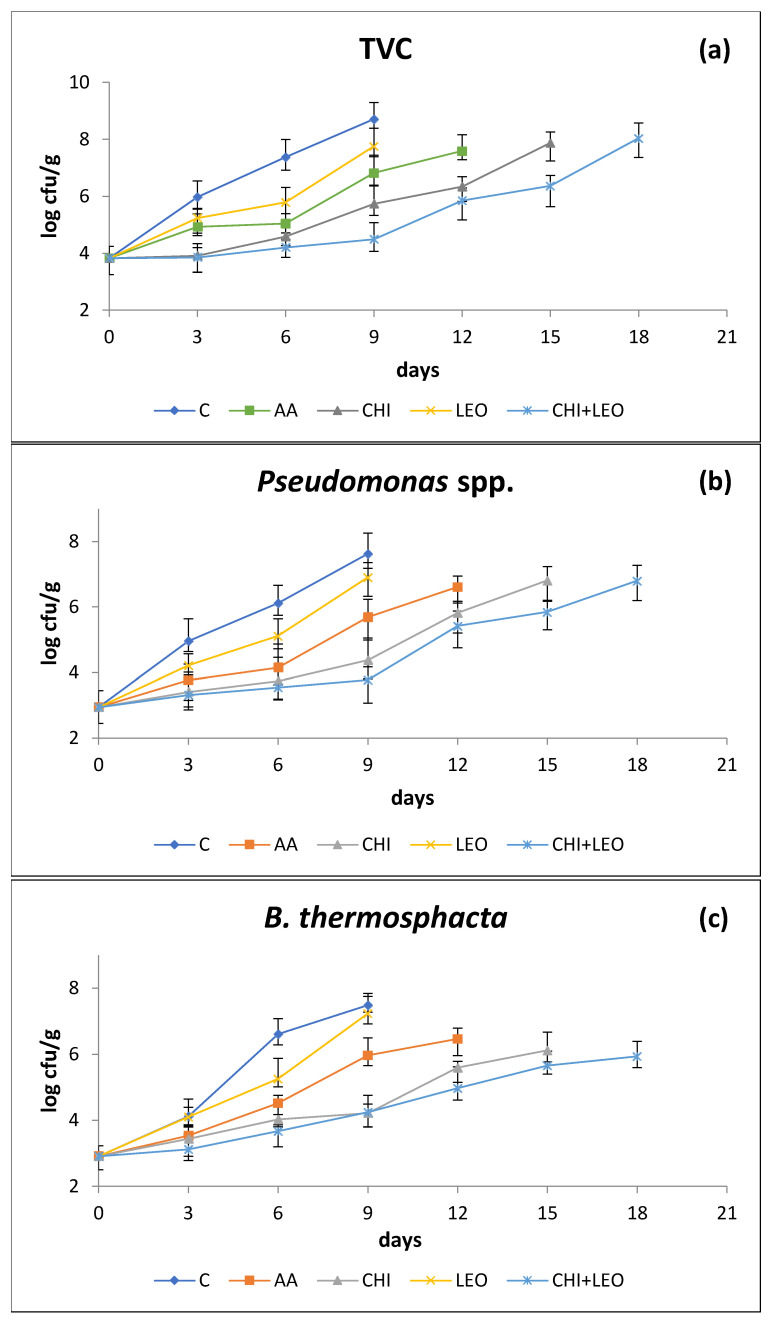
Combined effect of laurel essential oil and chitosan on (**a**) TVC, (**b**) *Pseudomonas* spp. counts, (**c**) *Brochothrix thermosphacta* counts, (**d**) Enterobacteriaceae counts, and (**e**) LAB counts of buffalo meat as a function of storage time (AA = acetic acid; CHI = chitosan; LEO = laurel essential oil).

**Figure 4 foods-11-01664-f004:**
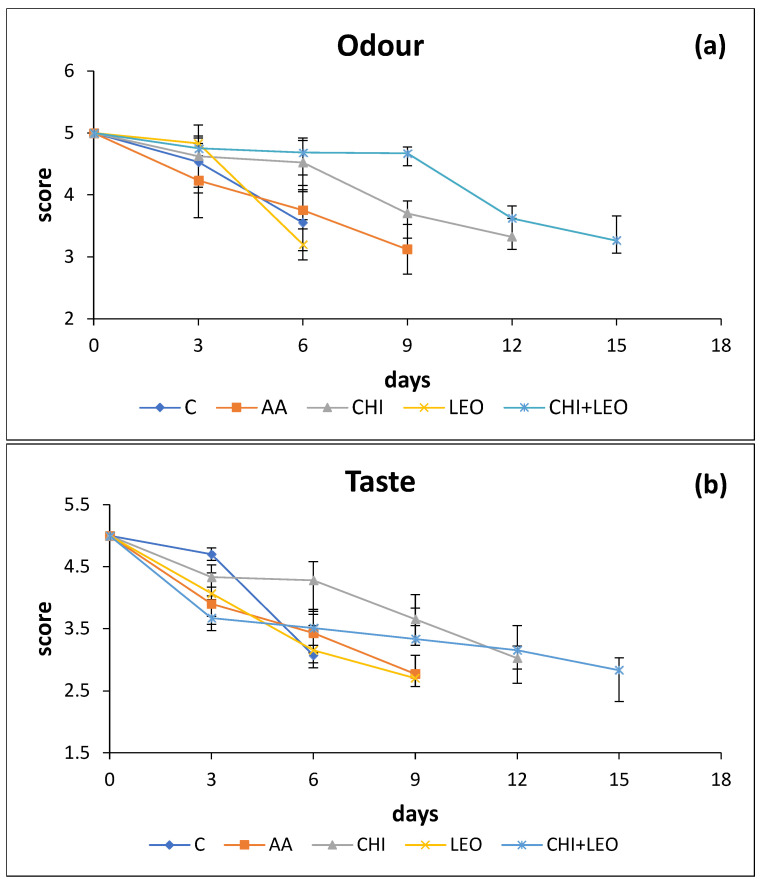
Combined effect of laurel essential oil and chitosan on (**a**) odour and (**b**) taste of cooked buffalo meat as a function of storage time (C = control; AA = acetic acid; CHI = chitosan; LEO = laurel essential oil).

**Table 1 foods-11-01664-t001:** Composition of laurel essential oil.

RT (min)	Library/ID	% Composition	RT (min)	Library/ID	% Composition
6.44	3-hexen-1-ol, (Z)	0.40 ± 0.02	12.66	L-borneol	0.37 ± 0.01
7.56	.alpha.-thujene	0.61 ± 0.03	12.77	4-terpineol	4.36 ± 0.28
7.76	.alpha.-pinene	5.79 ± 0.21	13.10	.alpha. terpineol	5.40 ± 0.26
8.07	camphene	1.10 ± 0.60	13.62	nerol	0.23 ± 0.01
8.33	sabinene	6.90 ± 0.32	15.06	4-thujen-2.alpha.-yl acetate	0.20 ± 0.02
8.40	.beta.-myrcene	0.44 ± 0.02	15.84	(-)-bornyl acetate	1.58 ± 0.07
8.51	2-.beta.-pinene	4.93 ± 0.23	17.93	alpha.terpinenyl acetate	16.40 ± 0.84
8.88	.alpha.-fellandrene	0.31 ± 0.02	18.15	eugenol	3.25 ± 0.015
8.94	.delta.3-carene	0.39 ± 0.03	19.52	methyl eugenol	4.22 ± 0.20
9.05	.alpha. terpinene	0.60 ± 0.02	20.97	caryophyllene	0.51 ± 0.03
9.18	Benzene, 1-methyl-2-(1-methylethyl)-	1.31 ± 0.07	21.20	trans-cinnamyl acetate	0.57 ± 0.04
9.29	dl-limonene	2.69 ± 0.12	23.73	.delta.-cadinene	0.43 ± 0.02
9.44	1,8-cineole	24.44 ± 1.26	25.69	(+) spathulenol	1.50 ± 0.08
9.77	.gamma.-terpinene	1.03 ± 0.04	25.95	caryophyllene oxide	1.41 ± 0.06
10.07	trans-sabinene hydrate	0.84 ± 0.08	26.39	.beta.-Ionone	0.34 ± 0.02
10.36	.alpha.-terpinolene	0.30 ± 0.02	26.68	E,E-.alpha.-farnesene	0.54 ± 0.04
10.44	L-linalool	2.99 ± 0.16	27.10	.gamma.-himachalene	0.45 ± 0.03
10.75	cis-sabinene hydrate	0.75 ± 0.05	27.34	alpha.amorphene	1.19 ± 0.06
11.71	1-terpineol	0.27 ± 0.02	27.69	.alpha.-cadinol	0.65 ± 0.04
11.88	trans-pinocarveol	0.31 ± 0.03			

**Table 2 foods-11-01664-t002:** Correlation between microbiological, physicochemical, and sensory attributes of buffalo meat on day 6 of refrigerated storage.

Pearson Correlation												
	TVC	*Pseudomonas* spp.	Enterobacte- Riaceae	LAB	*B. thermosphacta*	pH	TBA	Odour	Taste	L*	a*	b*
TVC	1											
*Pseudomonas* spp.	0.990 ^++^	1										
Enterobacteriaceae	0.968 ^++^	0.950 ^+^	1									
LAB	0.879 ^+^	0.905 ^+^	0.929 ^+^	1								
*B. thermosphacta*	0.999 ^++^	0.994 ^++^	0.972 ^++^	0.899 ^+^	1							
pH	0.649	0.657	0.439	0.296	0.629	1						
TBA	0.969 ^++^	0.952 ^+^	0.992 ^++^	0.901 ^+^	0.972 ^++^	0.480	1					
Odour	−0.937 ^+^	−0.895 ^+^	−0.803	−0.960 ^++^	−0.767	−0.155	−0.780	1				
Taste	−0.868 ^+^	−0.922 ^+^	−0.672	−0.690	−0.684	−0.377	−0.740	0.737	1			
L*	0.327	0.292	0.528	0.603	0.347	−0.458	0.444	−0.568	−0.044	1		
a*	0.071	0.209	0.038	0.334	0.107	0.069	0.058	−0.555	−0.557	−0.153	1	
b*	0.752	0.753	0.750	0.622	0.753	0.480	0.828	−0.568	−0.918 ^+^	0.039	0.191	1

^++^ Correlation is significant at the 0.01 level (2-tailed). ^+^ Correlation is significant at the 0.05 level (2-tailed).

## Data Availability

The data presented in this study are available on request from the corresponding author.

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
