# Peer review of "Combined Effect of Chitosan Coating and Laurel Essential Oil (Laurus nobilis) on the Microbiological, Chemical, and Sensory Attributes of Water Buffalo Meat"

_foods, 2022, doi:10.3390/foods11111664_

Round 1
Reviewer 1 Report
In this manuscript, Lydia et al. envestigated the combined effect of chitosan coating (CHI) and laurel essential oil (LEO) on the microbiological, chemical, and sensory attributes of water buffalo meat. I was glad to review your paper, however I would like to suggest the following points to strengthen your paper:
The authors should give more details about the panellist (such as age, gender,...). In the statistical analysis section the authors should start by saying why they perform an ANOVA, what is the goal? They also need to give information about the assumptions of ANOVA ( normality and homocedasticity). In this case, since the study involves more than one variable the authors should start by performing MANOVA followed by ANOVA´s and post hoc test, if possible.
In the results and Discussion section the authors should presented, for each one of the groups, the mean and standard deviation values
It is confusing for the reader to identify which groups are significantly different. Please identiffy these groups and give information about the p value. Also the figures legends should provide more information.
In the Results and Discussion section Authors mention: control samples, samples treated with (AA), samples treated with (LEO), samples treated with (CHI) and (CHI+LEO). The acronyms should be in the figure legends. It would have been a very useful for the reader.
Regarding to sensory analysis the authors should give information about the scale and about the information given to the pannelist.
In general, I recommend that the authors provide the results on which they were based to draw conclusions, as the graphical representation is not enough. I would greatly recommend the authors the put as much information as possible regarding to statistical analysis. Otherwise is difficult to understand the conclusions.
Line 195: ‘…. with statistically significant differences (p > 0.05) between the two until day 9 of storage.. This statement is opposite of the p-value given. Please correct .
Lines 196-205: At……..Results of the present study regarding pH are in…” Where is the data for this statement?
Line 218: “…reaching levels of 2.12 for control samples, 1.33 for samples treated with 218 (AA), 1.37 for samples treated with (LEO), 0.74 and 0.69 for samples treated with (CHI) 219 and (CHI+LEO) respectively, on day 6 of storage. This values are mean values? Please clarify.
Lines 224-226” …(LEO) signif-224 icantly affected (p < 0.05) the oxidation of buffalo meat. Chitosan was even more effective 225 in reducing the MDA content of buffalo meat by 2-fold during storage. Where are the results that support this statement?
Line 246: The authors mention “The decrease in L* values was significantly steeper in control samples.” Please give information the p-values obtained.
Line 247: The authors mention “On the other hand, both (LEO) and (CHI) provided partial protection to the colour (parameter L*) of buffalo meat exhibiting a lower reduction in meat sample lightness. Please clarify.
Line 380 : “…On the other hand, Chounou et 379 al. [12] reported a LAB reduction by only 0.2 log cfu/g (p > 0.05) on day 6 of storage of ground meat after the addition of 1% w/w chitosan. Please include “statistically non-significant “
Line 396: “ Odour and taste scores decreased significantly (p < 0.05) with storage time.” Can the authors be more specific? There were significant diferences between all the groups? How can we see that?
Line 397: The authors mention “ Taste proved to be a mor sensitive sensory attribute than odour.” Where is the data for this statement?
The conclusion section can be improved.
Author Response
All changes in the text are marked in red. We respond to the reviewer’s comments as follows:
Reviewer #1
I would like to suggest the following points to strengthen your paper:
Comment: The authors should give more details about the panellist (such as age, gender,...).
Response: We have added the following details on the panelists used in the study: “Panelists were 28 females and 23 males of the age group 22-60 who consume meat on a regular basis”. See revised text l. 175-179.
Comment: In the statistical analysis section the authors should start by saying why they perform an ANOVA, what is the goal? They also need to give information about the assumptions of ANOVA (normality and homocedasticity). In this case, since the study involves more than one variable the authors should start by performing MANOVA followed by ANOVA´s and post hoc test, if possible.
Response:
A two-way ANOVA is designed to assess the interrelationship of two independent variables on dependent variables. In the present study independent variables were different treatments and time. Dependent variables were pH, TBA, colour parameters: L*, a*, b*, odour, taste, TVC, Pseudomonads, LAB, Br. thermosphacta and Enterobacteriaceae. Thus, two-way ANOVA is adequate for the statistical treatments of data collected in this type of study.
Assumptions and limitations of a two-way ANOVA
(i) Dependent variables should be continuous – that is, measured on a scale which can be subdivided using increments (i.e. grams, milligrams)
(ii) The two independent variables should be in categorical, independent groups.
(iii) Sample independence – that is each sample has been drawn independently of the other samples
(iv) Variance Equality – That is the variance of data in the different groups should be the same
(v) Normality – That is each sample is taken from a normally distributed population
Comment: In the results and Discussion section the authors should presented, for each one of the groups, the mean and standard deviation values
Response: In Figs. 1-4 data is presented as mean ± standard error. In Table 1 we have added standard deviation values to % content of each volatile compound.
Comment: It is confusing for the reader to identify which groups are significantly different. Please identify these groups and give information about the p value. Also, the figures legends should provide more information.
Response: Statistically significant differences among treatments are given in terms of p values. Please see the text referring to p values in yellow highlighting. Given this information in the text, there is no need to repeat this in Figures.
Comment: In the Results and Discussion section Authors mention: control samples, samples treated with (AA), samples treated with (LEO), samples treated with (CHI) and (CHI+LEO). The acronyms should be in the figure legends. It would have been a very useful for the reader.
Response: Acronyms have been added to each Figure.
Comment: Regarding to sensory analysis the authors should give information about the scale and about the information given to the panelist.
Response: The scoring scale was 1-5 where 5 corresponded to the most liked sample and 1 corresponded to the least liked sample; a score of 3 was the lower acceptability limit. Minimum information was given to panelists with regard to the odour and taste of cooked meat. See revised text l. 175-179.
Comment: In general, I recommend that the authors provide the results on which they were based to draw conclusions, as the graphical representation is not enough.
Response: Conclusions drawn were based primarily on microbiological data (TVC upper limit of 7 log cfu/g) and sensory data (lower acceptability limit of 3). This piece of information has been added to the Conclusion section. See revised text l. 492-493.
Comment: I would greatly recommend the authors the put as much information as possible regarding to statistical analysis. Otherwise is difficult to understand the conclusions.
Response: Substantially more detailed information has been added to the Statistical analysis section See revised text l. 187-204.
Comment: Line 195: ‘…. with statistically significant differences (p > 0.05) between the two until day 9 of storage. This statement is opposite of the p-value given. Please correct
Response: The reviewer is right. We have corrected this error. See revised text l. 228.
Comment: Lines 196-205: At……..Results of the present study regarding pH are in…” Where is the data for this statement?
Response: pH data of the present study are presented in the text and in Fig 1a. These compare well to those of Samani et al. (pH ranging between 5.95 and 6.43). Please read the text.
Comment: Line 218: “…reaching levels of 2.12 for control samples, 1.33 for samples treated with 218 (AA), 1.37 for samples treated with (LEO), 0.74 and 0.69 for samples treated with (CHI) 219 and (CHI+LEO) respectively, on day 6 of storage. This values are mean values? Please clarify.
Response: Yes, these are mean values. See revised text l. 251.
Comment: Lines 224-226” …(LEO) significantly affected (p < 0.05) the oxidation of buffalo meat. Chitosan was even more effective 225 in reducing the MDA content of buffalo meat by 2-fold during storage. Where are the results that support this statement?
Response: Data on both the effect of LEO and CHI are presented in Fig. 1b.
Comment: Line 246: The authors mention “The decrease in L* values was significantly steeper in control samples.” Please give information the p-values obtained.
Response: See revised text l. 282-284.
Comment: Line 247: The authors mention “On the other hand, both (LEO) and (CHI) provided partial protection to the colour (parameter L*) of buffalo meat exhibiting a lower reduction in meat sample lightness. Please clarify.
Response: The sentence has been modified for clarification purposes. See revised text l. 286.
Comment: Line 380 : “…On the other hand, Chounou et al. [12] reported a LAB reduction by only 0.2 log cfu/g (p > 0.05) on day 6 of storage of ground meat after the addition of 1% w/w chitosan. Please include “statistically non-significant “
Response: Done. See revised text l. 416.
Comment: Line 396: “ Odour and taste scores decreased significantly (p < 0.05) with storage time.” Can the authors be more specific? There were significant diferences between all the groups? How can we see that?
Response: The sentence on odour and taste scores has been modified to become more specific. See revised text l. 433-434. Significant differences among different treatments were clearly observed after day 6 of storage. See revised text l. 435-437.
Comment: Line 397: The authors mention “ Taste proved to be a more sensitive sensory attribute than odour.” Where is the data for this statement?
Response: Data are presented in Fig. 4a and 4b. Also, a sentence has been added to the text. See revised text l. 435-437.
Comment: The conclusion section can be improved.
Response: We have added additional information to improve the Conclusion section. See revised text l. 496-498.
Reviewer 2 Report
In my opinion, the subject of this article is very interesting raw meat (water buffalo meat). The use of chitosan coatings and essential oils also seems to be very promising.
My comments on the content of the article:
- The title should be reworded (now: "Combined effect of chitosan coating and laurel essential oil (Laurus nobilis) on the microbiological, chemical, and sensory attributes of water buffalo meat"). The content of the article shows that it is closer to the content: "Effect of chitosan coating with laurel essential oil (Laurus nobilis) on the microbiological, chemical, and sensory attributes of water buffalo meat". What do you think about this change?
- Authors should use the same spelling throughout the text. For example, line 83 says "gram(+)" and "gram(-)", line 33 says "Gram-negative", and line 346 says "Gram-positive". It is good if the same writing methods are used throughout the text.
- Methodology:
- lines 111-115: I wonder if this method of applying the coating to the meat sample ensured that it contained exactly as much chitosan or mixture as was assumed? Were the samples weighed before and after the application of the substance?
- line 119: I don't quite understand the phrase "the second comprised samples dipped in a 1% v / v (AA) solution". If we find that the text in parentheses is not there, what will solution be? Please clarify this. - lines 144-145: please indicate in this paragraph the most important information regarding the method of identifying the volatile compounds. - line 149: I am puzzled by the indication of "225 mL of sterile Buffered Peptone Water solution" as a diluent for meat samples. Typically this volume of medium is used for pre-propagation of Salmonella bacteria. Isn't there a mistake here and shouldn't it be "Peptone Water solution"?
- lines 152-154: why were surface inoculations done? Most of the methods used in microbiology indicate deep inoculations. Please explain it. - Discuss the results:
- line 203: what does "the net reduction of pH" mean?
- graphs: I think that all graphs should start with "0" on both the X axis and the Y axis.
- cont. graphs: description of the assay variants - what do the variants "control + chitosan" or "control + laurel" mean? Either the sample is a control or some other variant - please change the determinations to those indicated in the methodology (chapter "Preparation of coating solutions and treatment of meat samples").
- microbiology graphs: why don't all lines has end at the same point (day 18)? Were no determinations made for some meat samples from day 9 onwards? I am asking for an explanation of this phenomenon. - Figure 3: why Authors use "counts" once and "population" in another?
- Conclusions: Please indicate in this paragraph the best solution, not a summary of the results. What was the best option about extending the shelf life?
- References: the items in lists 2, 6 and 17 have to little indications that would help the reader find them. Therefore, I am asking for additional information.
Author Response
All changes in the text are marked in red. We respond to the reviewer’s comments as follows:
Reviewer #2
My comments on the content of the article:
- The title should be reworded (now: "Combined effect of chitosan coating and laurel essential oil (Laurus nobilis) on the microbiological, chemical, and sensory attributes of water buffalo meat"). The content of the article shows that it is closer to the content: "Effect of chitosan coating with laurel essential oil (Laurus nobilis) on the microbiological, chemical, and sensory attributes of water buffalo meat". What do you think about this change?
Response: The present and the proposed by the reviewer title are essentially the same. They differ only in the word ‘combined’. We have examined separately the effect of CHI and LEO as well as the combination of these two. Thus, we have retained the title as is.
Authors should use the same spelling throughout the text. For example, line 83 says "gram(+)" and "gram(-)", line 33 says "Gram-negative", and line 346 says "Gram-positive". It is good if the same writing methods are used throughout the text.
Response: We have made changes as proposed by the reviewer. See revised text l. 85-86.
- Methodology:
- lines 111-115: I wonder if this method of applying the coating to the meat sample ensured that it contained exactly as much chitosan or mixture as was assumed? Were the samples weighed before and after the application of the substance?
Response: No, the samples were not weighed before and after the coating process. The calculation of 0.1 % of either (CHI) or (LEO) in meat was based on volume change of (CHI) and (LEO) solution before and after dipping of meat rather than on weight change. Differences between volume and weight changes should not be great.
- line 119: I don't quite understand the phrase "the second comprised samples dipped in a 1% v / v (AA) solution". If we find that the text in parentheses is not there, what will solution be? Please clarify this.
Response: the text has been slightly modified to become more clear. See revised text l. 121-124.
- - lines 144-145: please indicate in this paragraph the most important information regarding the method of identifying the volatile compounds.
Response: The methodology details are given. See revised text l. 152-157.
- - line 149: I am puzzled by the indication of "225 mL of sterile Buffered Peptone Water solution" as a diluent for meat samples. Typically this volume of medium is used for pre-propagation of Salmonella Isn't there a mistake here and shouldn't it be "Peptone Water solution"?
Response: buffered peptone water (BPW) solution was used. This is standard procedure according to APHA. We have deleted the word “sterile”.
- lines 152-154: why were surface inoculations done? Most of the methods used in microbiology indicate deep inoculations. Please explain it.
Response: For the determination of TVC, Pseudomonas spp., Brochothrix thermosphacta, Lactic acid bacteria (LAB) and Enterobacteriaceae the surface spreading plate procedure is used as these microorganisms are aerobic or facultative anaerobic.
Discuss the results:
- - line 203: what does "the net reduction of pH" mean?
Response: “net reduction of pH” means the result of two different opposing phenomena occurring simultaneously: i) the production of lactic acid through growth LAB resulting in a decrease of pH values and ii) protein breakdown for the production of alkaline compounds (NH3) resulting in increased pH values.
- - graphs: I think that all graphs should start with "0" on both the X axis and the Y axis.
Response: The X axis (Time axis) in all Figures starts with “0”. The Y axis does not necessarily start with ‘0’ but rather covers a different range of values in each case so as to clearly show differences between different treatments. Actually, the Y axis values are a blow up of what they would have been if the Y axis started from “0”. Imagine a Y axis for pH between 0 and 14. All the treatments having a pH around 6 would all super impose.
- cont. graphs: description of the assay variants - what do the variants "control + chitosan" or "control + laurel" mean? Either the sample is a control or some other variant - please change the determinations to those indicated in the methodology (chapter "Preparation of coating solutions and treatment of meat samples").
Response: The proposed changes have been made in Figures.
- - microbiology graphs: why don't all lines has end at the same point (day 18)? Were no determinations made for some meat samples from day 9 onwards? I am asking for an explanation of this phenomenon.
Response: Once the TVC of a given treatment exceeded the limit value of 7 log cfu/g, indicating the end of product shelf life/rejection of the sample, further analysis of the particular sample was discontinued.
Figure 3: why Authors use "counts" once and "population" in another?
Response: “populations” and “counts” are synonyms. The term “population” has been changed to ‘counts’ in the legend of Fig. 3.
- Conclusions: Please indicate in this paragraph the best solution, not a summary of the results. What was the best option about extending the shelf life?
Response: A sentence has been added to the Conclusion section indicating the best solution in terms of product shelf life extension. See revised text l. 496-498.
References: the items in lists 2, 6 and 17 have to little indications that would help the reader find them. Therefore, I am asking for additional information.
Response: References 2, 6 and 17 have been completed. See Reference list.
Reviewer 3 Report
Dear Authors,
The article entitled ”Combined effect of chitosan coating and laurel essential oil (Laurus nobilis) on the microbiological, chemical, and sensory attributes of water buffalo meat” is well written and present an interesting topic regarding the combined effect of chitosan coating (CHI) and laurel essential oil (LEO) on shelf-life extension of water buffalo meat stored under aerobic packaging conditions at 4oC.
Lines 21: Please verify, ca. 5-6 days …., probably approximatively 5-6 days ….
I recommend authors to complete the manuscript with a Principal Component Analysis in order to revealed the relationships between sensory characteristics and the other evaluated characteristics.
Please succinct describe the TBA method and the method regarding the identification of the volatile compounds.
The novelty and contribution to the area of interest should be stressed and substantiated.
Author Response
All changes in the text are marked in red. We respond to the reviewer’s comments as follows:
Reviewer #3
Comment: Lines 21: Please verify, ca. 5-6 days …., probably approximatively 5-6 days ….
Response: “ca.” has been changed to “approximately”. See revised text l. 21
Comment: I recommend authors to complete the manuscript with a Principal Component Analysis in order to revealed the relationships between sensory characteristics and the other evaluated characteristics.
Response: In order to satisfy the reviewer regarding the relationship between sensory characteristics and microbiological/physicochemical parameters determined, we ran a Pearson correlation the results of which are shown in Table 2 and l. 476-489 of the revised text.
Comment: Please succinct describe the TBA method and the method regarding the identification of the volatile compounds.
Response: Both methods (for the determination of TBA and volatiles) have been briefly described. See revised text l. 144-149 and 152-157.
Comment: The novelty and contribution to the area of interest should be stressed and substantiated.
Response: The novelty of the study has been stressed in a sentence added to the text. See revised text l. 90-91.
Round 2
Reviewer 1 Report
I recommend that the manuscript be accepted
Reviewer 3 Report
The manuscript can be accept in present form.